# Self-Reported Measures of Periodontitis in a Portuguese Population: A Validation Study

**DOI:** 10.3390/jpm12081315

**Published:** 2022-08-14

**Authors:** Vanessa Machado, Patrícia Lyra, Catarina Santos, Luís Proença, José João Mendes, João Botelho

**Affiliations:** 1Clinical Research Unit, Centro de Investigação Interdisciplinar Egas Moniz (CiiEM), Egas Moniz—Cooperativa de Ensino Superior, 2829-511 Almada, Portugal,; 2Evidence-Based Hub, CiiEM, Egas Moniz—Cooperativa de Ensino Superior, 2829-511 Almada, Portugal; 3Quantitative Methods for Health Research, CiiEM, Egas Moniz—Cooperativa de Ensino Superior, 2829-511 Almada, Portugal

**Keywords:** oral health surveys, periodontitis, periodontal disease, surveillance, precision, self-reported measures

## Abstract

Self-reported questionnaires have been developed and validated in multiple populations as useful tools to estimate the prevalence of periodontitis in epidemiological settings. This study aimed to explore the accuracy of self-reporting for predicting the prevalence of periodontitis in a Portuguese population. The questionnaires were given to patients at a university clinic. Thirteen self-reported questions on periodontal health were gathered in a patient-reported questionnaire. Then, self-reporting responses were validated using full-mouth periodontal examination as a comparison. Multivariable logistic regression was used to analyze sensitivity, specificity, accuracy, precision, and area under the curve-receiver operator characteristic (AUC-ROC). Self-reported answers from 103 participants (58 females and 45 males) were included. Self-reported gum health, loose teeth, tooth appearance, and use of dental floss were associated with different definitions of severe periodontitis. The self-reported questions on “having gum disease,” combined with “having gum treatment” and “having lost bone” were the items with higher performance for the 2018 case definition and the 2012 case definition, as well as for each respective severity staging. Categorization of tooth loss was only valuable for the prediction of periodontitis cases according to the 2012 case definition and its severe stage. Multiple self-reporting set-ups showed elevated performance levels for predicting periodontitis in Portuguese patients. These results may pave the way for future epidemiological surveillance programs using self-reporting approaches.

## 1. Introduction

Periodontal diseases, including gingivitis and periodontitis, are the sixth-most prevalent conditions contributing to the global burden of chronic non-communicable illnesses [1]. Beyond its prevalence, affecting more than 50% of the adult population worldwide [1], periodontal disease has been shown to have implications for quality of life [2] and nutrition [3], aesthetic confidence, and general well-being, varying with the increasing grading and severity of periodontitis [2,4]. In addition to its social impact, periodontitis is robustly linked with systemic disease and inflammatory burden [5].

Clinically-based periodontal examinations, including full-mouth periodontal probing depth (PD), clinical attachment loss (CAL), and bleeding on probing (BOP), are the gold standard for periodontitis diagnosis [6]. These clinical measures are not complex to obtain but are time-consuming, require trained professionals, specialized equipment, and infection control strategies. Facing these challenges, the Centers for Disease Control and Prevention (CDC) proposed a self-reported strategy to screen periodontitis that has been successfully validated. Particularly in screening systems and epidemiological research, it becomes highly relevant to explore such non-clinical and self-reported approaches to make periodontal conditions surveillance more cost-effective [7].

The eight-item questionnaire proposed in a collaboration between CDC and the American Academy of Periodontology (AAP) was validated and implemented in the National Health and Nutrition Examination Survey (NHANES) [8]. This tool is a valid instrument to screen the prevalence of periodontitis in American dentate adults [9,10,11]. This CDC/AAP questionnaire has been successfully cross-culturally adapted and validated for Spanish [12,13], French [14], Brazilian [15], Dutch [16] and Japanese [17] populations, with discriminative capability for detecting individuals with periodontitis. Nonetheless, this epidemiological tool has never been validated in Portugal, a country with high prevalence and burden of periodontitis.

The present study aimed to cross-culturally adapt and validate a periodontitis self-reported questionnaire based on the European Federation of Periodontology (EFP)/AAP and CDC/AAP tool together with other questions with predictive performance in a Portuguese population. To our view, assessing the validity of translations of the self-reported questions to Portuguese and attesting its predictive capacity will be an important addition to the literature on the surveillance of periodontitis.

## 2. Materials and Methods

This study is reported following the Transparent Reporting of a multivariable prediction model for Individual Prognosis or Diagnosis (TRIPOD) guidelines for reporting predictive models [18]. This study received approval by the institutional review board (Egas Moniz Ethics Committee nº. 595) and we obtained signed informed consent from all participants. This study was developed in compliance with the World Medical Association Helsinki Declaration (2013).

### 2.1. Study Design and Participants

First-time patients seeking dental care at the Egas Moniz Dental Clinic (EMDC), located in Almada (Portugal), were invited to participate in the study. Patients were deemed eligible if they met the following inclusion criteria: aged at least 18 years old; being able to read and understand Portuguese; agreeing to participate in the study; completing the questionnaire; and receiving a complete periodontal examination. As exclusion criteria, patients having less than two teeth or who had been pre-diagnosed with a severe or terminal disease (for instance, advanced heart failure, end-stage kidney disease, or advanced-stage cancer) [17] were not included.

Sample size calculation was conducted in R (version 4.0.0) using the ‘pwr’ package. Considering a prevalence of periodontitis of 59.9%, observed in a previous large epidemiological study undertaken in the Almada-Seixal region (Portugal) [19], we estimated a minimum required sample of 101 patients, considering a 5% significance level and 80% power.

### 2.2. Self-Reported Questions

The questionnaire included a total of thirteen self-reported questions, with eight of them based on the original CDC/AAP [8]. The remaining five questions were selected from a group of past studies that presented good performance results [13,17] (Table 1).

### 2.3. Study Design and Participants

The questionnaire collected information regarding age, sex, smoking status, education level, and toothbrushing per day. Information regarding data management for the logistic regression analysis can be found in Section 2.5 (Statistical analysis).

### 2.4. Periodontal Examination and Periodontitis Case Definition

Following the self-reported questionnaire, patients underwent periodontal examination by trained and calibrated examiners (kappa values > 0.8) via a periodontal probe (PCP 12, Hu-Friedy Chicago, IL, USA). The examiners were blinded to the questionnaire responses. A circumferential, six sites per tooth examination protocol was performed, excluding third molars, implants, and retained roots. PD, CAL and gingival margin recession (REC) were registered to the nearest mm. BOP was also recorded. Tooth loss was registered as well, accounting for the total number of teeth present. Data was uploaded to a Microsoft Excel database and used to compute periodontal case definitions using algorithms as previously described [20,21].

Cases were diagnosed as periodontitis and according to each staging based on the EFP/AAP 2018 case definition and stage III/IV form (accepted as severe stage) [6]:

Periodontitis case: Interdental CAL at ≥2 non-adjacent teeth, or buccal/oral CAL ≥ 3 mm with pocketing > 3 mm at ≥2 teeth;Severe periodontitis: interdental CAL at site of greatest loss of ≥5 mm; or, radiographic bone loss extending to mid-third of root and beyond.

To explore the suitability of self-reported measures to a wide variety of periodontitis case definitions we also defined patients according to The CDC/AAP 2012 [22]:

Periodontitis (using the mild definition as cutoff): ≥2 interproximal sites with CAL ≥ 3 mm, and ≥2 interproximal sites with PPD ≥ 4 mm (not on same tooth) or one site with PD ≥ 5 mm;Severe periodontitis: ≥2 interproximal sites with CAL ≥ 6 mm (not on the same tooth) and ≥1 interproximal site with PPD ≥ 5 mm.

### 2.5. Statistical Analysis

All analyses were conducted in R (version. 4.0.0), using the package ‘cvAUC’. Age was dichotomized to <45 (0) and ≥45 years old (1) according to a previous study report) [19]. Smoking status was dichotomized into no active smoking (0) (including non-smokers and former smokers) or active smoking (1). Education was categorized as: elementary school (0) or middle school/higher education (1). Tooth loss was obtained from the periodontal examination (Section 2.4) and was categorized as 0, 1–5 or ≥6 tooth loss (for logistic regression this was dichotomized as <6 teeth lost (0) or ≥6 (1)).

Multivariable logistic regression analyses were used to predict the periodontal outcomes, according to the different case definitions that were tested. Crude and adjusted odds ratio (OR) were determined within these procedures.

Four sets of predictor variables were tested against these periodontal outcomes using multivariable binary logistic regression procedures: (a) the thirteen oral health self-reported questions; (b) demographic and risk factors (age, sex, educational level, smoking status, diabetes, and tooth loss); (c) combined self-reported oral health questions and demographic/risk factors; and (d) the selection of the best significant subset of predictor variables using the method of all possible equations. The significance level was set at 5%.

The predictive validity of these questions for the different disease definitions was assessed by the area under the curve-receiver operator characteristic (AUC/ROC) analysis. Sensitivity, specificity, accuracy, precision, and the Akaike Information Criterion (AIC) were evaluated.

## 3. Results

### 3.1. Participants’ Characteristics

All participants were recruited between November 2021 and April 2022. In this population subset, 61.2% and 68.9% were diagnosed with periodontitis, according to the EFP/AAP 2018 and CDC/AAP 2012 periodontitis case definitions, respectively. The mean age of the total sample was 50.6 (±16.0) years, with the most severe periodontitis being diagnosed in older participants: 59.2 (±11.5) and 58.3 (±11.7). Those participants with severe periodontitis were mainly males (56.4% and 55.4%), presenting lower education levels (41.0% and 42.9%), predominantly non-smokers (51.3% and 48.2%), and had six or more teeth missing (69.2% and 67.9%), according to EFP/AAP 2018 and CDC/AAP 2012 case definitions, respectively (Table 2).

### 3.2. Predictive Ability of the Self-Reported Questions

The overall response distribution to the thirteen questions depicted a scarce percentage of ‘Refused/Don’t know’ responses, and a very substantial response rate in all questions (Appendix A). Each self-reported question showed significant association results with the different periodontitis definitions used (EFP/AAP 2018 and CDC/AAP 2012) (Table 3). A positive response to the question on ‘Gum disease’ (Q1) (54.4), was highly associated with having severe periodontitis according to EFP/AAP 2018; those patients unanimously (100%) indicated a positive response. Contrariwise, the questions about ‘Floss use’, ‘Gum bleeding last 3 months’, and ‘Gum bleeding’ were not associated with periodontitis (Table 3). The strongest crude associations were found for ‘Gum disease’, ‘Loose tooth’, ‘Lost bone’ and ‘Gum retraction’ in participants with severe periodontitis according to EFP/AAP 2018.

In multivariate modeling (Table 4 and Table 5), reduced models outperformed the prediction of periodontitis and severe periodontitis according to both the EFP/AAP definition and the CDC/AAP definition. All reduced models showed an AUC over 0.80, suggesting excellent discrimination potential for periodontitis (0.86), according to the EFP/AAP definition, and periodontitis (0.86) and its severe form (0.80) in the CDC/AAP definition. For severe periodontitis according to the EFP/AAP 2018 periodontal case definition, the prediction value was deemed acceptable (0.71). All reduced models included the question regarding recognizing having ‘Gum disease’ (Q1). Furthermore, ‘Had Gum Treatment’ (Q3)], ‘Loose tooth’ (Q4), ‘Lost bone’ (Q5), and ‘Loose Tooth Loss’ as a risk factors were also significant questions.

## 4. Discussion

In this study, the validity of a 13-item self-reported questionnaire for periodontitis, translated to Portuguese, was assessed. We tested the performance of this set of questions in predicting cases of both periodontitis and severe periodontitis in two accepted case definitions. This questionnaire included eight items of the original CDC/AAP questionnaire [8] and five items from other similar studies [13,17]. Our results show that the predictive values of the reduced models were 86% in both the EFP/AAP 2018 and the CDC/AAP 2012 case definitions, while for severe periodontitis the reduced models ranged between 71% and 80% for the EFP/AAP 2018 and the CDC/AAP 2012, respectively. In line with previous studies using similar methodology, these results may provide potential for periodontal surveillance and screening, but might not be useful for etiological studies [12].

The self-report questionnaire for periodontitis not only is a relevant alternative strategy to the clinical diagnosis, with good validity and a reliability to screen individuals with periodontitis, but it also requires fewer resources [8,11,23]. This method enables large scale epidemiological studies and facilitates the challenging diagnosis of periodontitis at the populational level [17]. In fact, self-report as a tool has been previously validated in other medical contexts as well [24,25,26]. Therefore, we believe this questionnaire is a particularly important tool for tracking and managing public health measures for oral health, particularly in Portugal.

In the Portuguese context, oral health care is mostly centralized in private practices using an out-of-pocket system and dental services are generally lacking in the National Health Service [27]. This limits the population’s access to professional dental services and is felt more keenly by deprived socioeconomic and disadvantaged people. Nonetheless, a system of “dental vouchers” was created in 2008 to promote free oral health care in target populations (children, adolescents, and vulnerable groups such as pregnant women, patients with Human Immunodeficiency Virus, and elders with low-socioeconomic status) [28]. This panorama might explain the high prevalence of periodontal disease in the Portuguese population, as reported recently by Botelho et al. [19], and previously [29], although the latter did not follow the gold standard full mouth periodontal examination, compromising the sensibility and specificity of the results [30]. The burden of periodontal disease in Portugal supports the need for a comprehensive national oral program, the implementation of simple but accurate screening tools for oral diseases (such as the self-report method) and the promotion of awareness campaigns for oral health.

Our results using multivariable prediction models, including the self-reported perception of gum disease and previous gum treatment with the number of teeth lost characteristics, performed well in predicting periodontitis cases (AUC = 0.86), for both periodontal classifications. Specifically in severe periodontitis, the gum disease and lost bone self-reported information had the highest predictive ability (AUC = 0.71 and 0.80 for EFP/AAP 2018 and CDC/AAP 2012, respectively). Comparing our findings with previous studies, the multivariable prediction model for severe periodontitis had an AUC of 0.83 [8], 0.82 [14], 0.75 [12], 0.86 [13], 0.82 [16] in the American (NHANES), French, Dutch and Spanish (both the Di@bet.es Study participants and in the Catalonian adult subjects) populations, respectively. The slightly different predictive capacity could be attributed to population characteristics or periodontal case definition. For example, Carra et al. [14] used a periodontal screening score while Iwasaki et al. [17] used the CDC/AAP 2012 periodontal case definition. 

From a surveillance point of view, the abundance of new strategies and tools seems to guide us towards a possible combination of technological strategies that will be very powerful in the preventive diagnosis of periodontitis. Combined with self-report questionnaires that are applicable in any setting (i.e., at any sort of health service or even at home), we will be able to create tree health and risk assessments along with 1.0 and 2.0 software tools [31,32]. We foresee that a set of interactions with patients may be carried out prior to any clinical diagnosis, based only on self-report and complementary diagnostic tests (radiographs, blood and/or salivary analyses), by using artificial intelligence empowered strategies [33]. This set of questions could constitute a real evolution in periodontal diagnosis and treatment towards a holistic, more personalized process, and one more focused on the individual needs of patients. In addition, it will allow a closer monitoring of patients treated and under maintenance programs, strengthening the prevention of periodontitis recurrence and its risk factors.

### Strengths and Limitations

The translation to Portuguese was performed by experts, following a rigorous but clear linguistic strategy, in order to keep the Portuguese version an equivalent measure of the original English questionnaire, while maintaining good content-related validity. This sample was a consecutive pool of newly incoming patients at the university dental clinic, although its location may have contributed to limit the generalizability of our results to the entire Portuguese population. In addition, we have employed widely used periodontitis case definitions and its severity staging, enhancing the importance of such results, and empowering its clinical validity in future screening/surveillance.

Furthermore, the accuracy and heterogeneity of results are influenced by other determinants that are intrinsically related to the populations studied, such as health literacy, access, socioeconomic status, or awareness of dental care services [12].

## 5. Conclusions

Self-report measures of periodontitis combined with risk factors showed predictive validity towards periodontitis and its severe form in the Portuguese population, using the 2018 and 2012 case definitions.

## Figures and Tables

**Table 1 jpm-12-01315-t001:** Original and Portuguese versions of the self-reported periodontal screening questionnaire.

Original Question	Translated Question in Portuguese	Question Variable	Reference
**Do you think you might have gum disease?** Yes|No|Don’t know|Refused	**Acha que pode ter doença das gengivas?** Sim|Não|Não Sei|Recuso-me a responder	Gum disease	Eke et al. (2013) [8]
**Overall, how would you rate the health of your teeth and gums?** Excellent/Very good/Good/Fair/Poor/Don’t know/Refused	**No geral, como classificaria a saúde de seus dentes e gengivas?** Excelente|Muito boa|Boa|Razoável|Fraca|Não Sei|Recuso-me a responder	Teeth/gum health	Eke et al. (2013) [8]
**Have you ever had treatment for gum disease, such as scaling and root planning, sometimes called “deep” cleaning?** Yes|No|Don’t know|Refused	**Alguma vez recebeu tratamento para a doença das gengivas tal como alisamento ou raspagem radicular, por vezes chamada limpeza profunda?** Sim|Não|Não Sei|Recuso-me a responder	Had gum treatment	Eke et al. (2013) [8]
**Have you ever had any teeth become loose on their own, without an injury?** Yes|No|Don’t know|Refused	**Já teve algum dente que começou a abanar por si só, sem nenhum trauma ou lesão?** Sim|Não|Não Sei|Recuso-me a responder	Loose tooth	Eke et al. (2013) [8]
**Have you ever been told by a dental professional that you lost bone around your teeth?** Yes|No|Don’t know|Refused	**Já algum profissional dentário lhe disse que tinha perdido osso ao redor dos dentes?** Sim|Não|Não Sei|Recuso-me a responder	Lost bone	Eke et al. (2013) [8]
**During the past 3 months have you noticed a tooth that doesn’t look right?** Yes|No|Don’t know|Refused	**Nos últimos 3 meses, reparou se algum dente não tenha estado bem?** Sim|Não|Não Sei|Recuso-me a responder	Tooth does not look right	Eke et al. (2013) [8]
**Aside from brushing your teeth with a toothbrush, in the last 7 days, how many times did you use dental floss or any other device to clean between your teeth?** _______ Number|Refused	**Além de escovar os dentes com escova de dentes, nos últimos sete dias, quantas vezes usou fio dentário ou qualquer outro dispositivo para limpar a região entre os dentes?** _______ Número|Recuso-me a responder	Floss use	Eke et al. (2013) [8]
**Do your gums usually bleed either when brushing or chewing?** Yes|No|Don’t know|Refused	**Sangra habitualmente das gengivas sem escovar ou mastigar?** Sim|Não|Não Sei|Recuso-me a responder	Gum bleeding	Saka-Herrán et al. (2020) [13]
**During the past three months, have you had bleeding gums?** Never|Hardly ever|Sometimes|Fairly often|Very often	**Nos últimos 3 meses, sangrou das gengivas?** Nunca|Quase nunca|Ás vezes|Com bastante frequência|Muitas vezes	Gum bleeding last 3 months	Iwasaki et al. (2021) [17]
**Have you lost teeth in recent years because of mobility?** Yes|No|Don’t know|Refused	**Perdeu algum dente nos últimos anos por ter começado a abanar?** Sim|Não|Não Sei|Recuso-me a responder	Loose teeth loss	Saka-Herrán et al. (2020) [13]
**Have you felt pain in your gums during the last months?** Yes|No|Don’t know|Refused	**Nos últimos meses, doeram-lhe as suas gengivas?** Sim|Não|Não Sei|Recuso-me a responder	Gum pain	Saka-Herrán et al. (2020) [13]
**In the past years have you noticed that your teeth are longer or that you have receding gums?** Yes|No|Don’t know|Refused	**Nos últimos meses, notou que os seus dentes estão maiores e as gengivas mais retraídas do que o normal?** Sim|Não|Não Sei|Recuso-me a responder	Gum retraction	Saka-Herrán et al. (2020) [13]
**In the last years have you noticed that you see the roots of several of your teeth?** Yes|No|Don’t know|Refused	**Nos últimos meses, notou que se vêem as raízes dos seus dentes?** Sim|Não|Não Sei|Recuso-me a responder	Roots visible	Saka-Herrán et al. (2020) [13]

**Table 2 jpm-12-01315-t002:** Socio-demographic characteristics of the participants.

Variable	Total (n = 103)	EFP/AAP 2018 Periodontitis (n = 63)	EFP/AAP 2018 Severe Periodontitis (n = 39)	CDC/AAP 2012 Periodontitis (n = 71)	CDC/AAP 2012 Severe Periodontitis (n = 56)
Age	50.6 (16.0)	56.3 (13.2)	59.2 (11.5)	54.7 (13.9)	58.3 (11.7)
**Sex**					
Female	58 (56.3)	31 (49.2)	17 (43.6)	38 (53.5)	25 (44.6)
Male	45 (43.7)	32 (50.8)	22 (56.4)	33 (46.5)	31 (55.4)
**Education level**					
Elementary	32 (31.1)	27 (42.9)	16 (41.0)	27 (38.0)	24 (42.9)
Middle	39 (37.9)	20 (31.7)	12 (30.8)	25 (35.2)	18 (32.1)
Higher	32 (31.1)	16 (25.4)	11 (28.2)	19 (26.8)	14 (25.0)
**Smoking habits**					
Non-smoker	63 (61.2)	33 (52.4)	20 (51.3)	38 (53.5)	27 (48.2)
Former smoker	13 (12.6)	11 (17.5)	7 (17.9)	12 (16.9)	11 (19.6)
Current smoker	27 (26.2)	19 (30.2)	12 (30.8)	21 (29.6)	18 (32.1)
**Tooth loss**					
0	21 (20.4)	4 (6.3.0)	0 (0.0)	5 (7.0)	2 (3.6)
1–5	36 (35.0)	19 (30.2)	12 (30.8)	24 (33.8)	16 (28.6)
≥6	46 (44.7)	40 (63.5)	27 (69.2)	42 (59.2)	38 (67.9)
**Toothbrushing per day**					
1	14 (13.6)	12 (19.0)	5 (12.8)	12 (16.9)	11 (19.6)
2	57 (55.3)	34 (54.0)	20 (51.3)	38 (53.5)	30 (53.6)
3 or more	32 (31.1)	17 (27.0)	14 (35.9)	21 (29.6)	15 (26.8)

Categorical variables are expressed as n (%), and continuous variables are expressed as mean (SD), within each periodontal condition category. n—number of participants.

**Table 3 jpm-12-01315-t003:** Adjusted odds ratio (OR) for each of the four periodontitis outcomes.

Question	Adjusted OR (Standard Error) ^a^			
	EFP/AAP 2018 Periodontitis (n = 63)	EFP/AAP 2018 Severe Periodontitis (n = 39)	CDC/AAP 2012 Periodontitis (n = 71)	CDC/AAP 2012 Severe Periodontitis (n = 56)
Gum disease	53.0 (8.5–330.9) ***	4.3 (1.8–10.1) ***	30.0 (5.8–156.0) ***	14.7 (3.7–58.1) ***
Teeth/gum health	7.7 (0.1–508.0)	-	109.8 (10.7–1126.8) ***	-
Had gum treatment	12.4 (2.4–63.2) **	1.9 (0.8–4.5) **	20.3 (2.4–171.8) **	6.2 (1.8–21.6) **
Loose tooth	13.6 (2.7–69.2) **	6.9 (2.8–17.1) ***	8.4 (2.4–30.1) *	5.6 (1.7–18.0) **
Lost bone	12.4 (2.9–54.1) ***	7.1 (2.9–17.4) ***	-	18.3 (4.3–77.9) ***
Tooth does not look right	4.4 (1.4–13.8) **	5.2 (2.1–12.8)	0.7 (0.3–1.7) *	2.9 (1.0–8.2) *
Floss use	1.3 (0.4–3.7)	0.7 (0.3–1.5)	1.8 (1.1–3.1)	1.9 (0.6–5.7)
Gum bleeding	6.2 (1.8–21.4) **	1.5 (0.7–3.3)	2.7 (0.9–7.7)	2.7 (0.9–8.1)
Gum bleeding last 3 months	6.0 (0.3–107.9)	2.9 (1.0–8.3)	-	2.6 (0.6–11.8)
Loose tooth loss	31.5 (1.8–561.8) *	3.9 (1.5–10.5) **	11.0 (1.1–111.9) *	7.3 (1.2–44.6) *
Gum pain	6.3 (1.4–27.9) *	1.2 (0.5–3.1)	2.8 (0.8–9.5)	2.3 (0.7–7.9)
Gum retraction	3.0 (0.9–9.9)	4.6 (1.9–11.0) ***	12.5 (2.5–63.3) **	3.7 (1.1–11.9) *
Roots visible	2.7 (0.8–9.2)	4.1 (1.7–10.0) **	9.3 (1.8–47.7) **	2.6 (0.8–8.4)

^a^ Multivariable logistic regression adjusted for age, sex, education, smoking habits, tooth loss, and employment status. * *p* < 0.05, ** *p* < 0.01, *** *p* < 0.001.

**Table 4 jpm-12-01315-t004:** Logistic regression models for the periodontitis EFP/AAP case definition of 2018 and respective performance analysis.

	EFP/AAP 2018 Periodontitis (Models)				EFP/AAP 2018 Severe Periodontitis (Models)			
Question	1	2	3	4	1	2	3	4
Gum disease	X		X	X	X		X	X
Teeth/gum health	X		X		X		X	
Had gum treatment	X		X	X	X		X	
Loose tooth	X		X		X		X	
Lost bone	X		X	X	X		X	X
Tooth does not look right	X		X		X		X	
Floss use	X		X		X		X	
Gum bleeding	X		X		X		X	
Gum bleeding last 3 months	X		X		X		X	
Loose tooth loss	X		X		X		X	
Gum pain	X		X		X		X	
Gum retraction	X		X		X		X	
Roots visible	X		X		X		X	
Age		X	X			X	X	
Sex		X	X			X	X	
Education		X	X			X	X	
Employment status		X	X			X	X	
Smoking habits		X	X			X	X	
AUC	0.58	0.49	0.50	0.86	0.55	0.50	0.50	0.71
Sensitivity (%)	100.0	0.98	100.0	88.9	100.0	100.0	100.0	79.5
Specificity (%)	15.4	0.0	0.0	82.5	9.4	0.0	0.0	62.5
Accuracy (%)	67.6	61.2	60.2	86.4	43.7	37.9	37.9	68.9
Precision (%)	65.6	61.2	60.8	88.9	40.2	37.9	37.9	66.0
AIC	78.3	113.1	75.8	87.0	116.2	119.8	130.1	117.8

AAP, American Academy of Periodontology; AIC, Akaike Information Criterion; AUC, Area Under the Curve; EFP, European Federation of Periodontology; P, Periodontitis; SP, Severe Periodontitis.

**Table 5 jpm-12-01315-t005:** Logistic regression models for the periodontitis CDC/AAP case definition of 2012 and respective performance analysis.

	EFP/AAP 2018 Periodontitis (Models)				EFP/AAP 2018 Severe Periodontitis (Models)			
Question	1	2	3	4	1	2	3	4
Gum disease	X		X	X	X		X	X
Teeth/gum health	X		X		X		X	
Had gum treatment	X		X	X	X		X	X
Loose tooth	X		X		X		X	
Lost bone	X		X		X		X	
Tooth does not look right	X		X		X		X	
Floss use	X		X		X		X	
Gum bleeding	X		X		X		X	
Gum bleeding last 3 months	X		X		X		X	
Loose tooth loss	X		X		X		X	
Gum pain	X		X		X		X	
Gum retraction	X		X		X		X	
Roots visible	X		X		X		X	
Age		X	X			X	X	
Sex		X	X			X	X	
Education		X	X			X	X	
Employment status		X	X			X	X	
Smoking habits		X	X			X	X	
Tooth Loss		X	X	X				X
AUC	0.57	0.49	0.50	0.86	0.56	0.50	0.49	0.80
Sensitivity (%)	98.6	98.4	100.0	96.8	100.0	100.0	98.2	96.8
Specificity (%)	15.6	0.0	0.0	75.0	12.8	0.0	0.0	75.0
Accuracy (%)	72.8	60.2	68.9	88.3	60.2	54.4	53.4	88.3
Precision (%)	72.2	60.8	68.9	85.9	57.7	54.4	53.9	85.9

AAP, American Academy of Periodontology; AIC, Akaike Information Criterion; AUC, Area Under the Curve; CDC, Centers for Disease Control and Prevention; P, Periodontitis; SP, Severe Periodontitis.

## Data Availability

The data of this study is available on request from the corresponding author, and not publicly available due to ethical and legal restrictions imposed by the Egas Moniz Ethics Committee.

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
