# Peer review of "Self-Reported Measures of Periodontitis in a Portuguese Population: A Validation Study"

_jpm, 2022, doi:10.3390/jpm12081315_

Round 1

Reviewer 1 Report

The authors aimed to validate a Portuguese questionnaire for "predicting" the prevalence of periodontitis.

1. The sample population is limited considering the small volume and their attributes. Patients "seeking dental care" are not appropriate for a questionnaire aiming at the general population. 

2. The performance of the Portuguese questionnaire should be discussed in comparison with versions of other languages and highlighted for potential applications or imporvement.

Author Response

Dear Editor of the Special Issue “Prevention and Management of Oral Healthcare” of Journal of Personalized Medicine, 

Dr. Chun-Ming Chen 

We are pleased with the opportunity to revise and resubmit our manuscript entitled "Self-Reported Measures of Periodontitis in a Portuguese Population: A Validation Study” JPM (ISSN 2075-4426).

The comments from the reviewers were important and thoroughly considered. According to your instructions, below you will find a point-by-point response to the comments along with our revision. All changes are shown in the manuscript text file with track changes.

We hope this revised manuscript will enable its further consideration. We are happy to consider further revisions and we thank you for your continued interest in our research.

REVIEWER COMMENTS

Reviewer #1 (Remarks to the Author):

The authors aimed to validate a Portuguese questionnaire for "predicting" the prevalence of periodontitis.

  1. The sample population is limited considering the small volume and their attributes. Patients "seeking dental care" are not appropriate for a questionnaire aiming at the general population. 

Our answer: While appreciating this commentary, we respectfully disagree on considering this sample population as small. We have followed a similar strategy conducted previously on studies of the same design, where they have estimated a minimum required sample from previous epidemiological studies (Saka-Herrán et al. 2020; Carra et al. 2018). In this case, we were lucky enough to have been the group responsible for conducting the most recent epidemiological study on the region. Concerning the fact patients seeking dental care are not appropriate we also disagree considering such patients were randomly included and the Egas Moniz Dental Clinic is a clinical facility open to all Lisbon Metropolitan Area with large representation from this area as previously detailed, and also previous studies employed the same strategy (Saka-Herrán et al. 2020; Carra et al. 2018; Cyrino et al. 2011).

  1. The performance of the Portuguese questionnaire should be discussed in comparison with versions of other languages and highlighted for potential applications or improvement.

Our answer: We have done this accordingly in the original submission in lines 233-245. We appreciate this remark.

References

Saka-Herrán C, Jané-Salas E, González-Navarro B, Estrugo-Devesa A, López-López J. Validity of a self-reported questionnaire for periodontitis in Spanish population. J Periodontol. 2020 Jan 26. doi: 10.1002/JPER.19-0604. Epub ahead of print. PMID: 31984491

Carra MC, Gueguen A, Thomas F, Pannier B, Caligiuri G, Steg PG, Zins M, Bouchard P. Self-report assessment of severe periodontitis: Periodontal screening score development. J Clin Periodontol. 2018 Jul;45(7):818-831. doi: 10.1111/jcpe.12899. Epub 2018 May 16. PMID: 29611224.

Cyrino RM, Miranda Cota LO, Pereira Lages EJ, Bastos Lages EM, Costa FO. Evaluation of self-reported measures for prediction of periodontitis in a sample of Brazilians. J Periodontol. 2011 Dec;82(12):1693-704. doi: 10.1902/jop.2011.110015. Epub 2011 May 12. PMID: 21563951.

Heaton B, Gordon NB, Garcia RI, Rosenberg L, Rich S, Fox MP, Cozier YC. A Clinical Validation of Self-Reported Periodontitis Among Participants in the Black Women's Health Study. J Periodontol. 2017 Jun;88(6):582-592. doi: 10.1902/jop.2017.160678. Epub 2017 Jan 15. PMID: 28088874; PMCID: PMC5556388.

Reviewer 2 Report

The study is useful and can contribute to the assessment of periodontal disease in a Portuguese population. The study has powerful statistics. please clarify the following points prior to further processing.

title: if validation is used in the title, then it should reflect in the study objective, otherwise your study is a "survey-based study"

Introduction:

Add significance of the study in the last paragraph.

methods:

The section should have a separate heading on questionnaire validation"

please clarify which methods were adopted to validate the questionnaire i.e., face validity, content validity, construct validity, or criterion validity.

The questionnaire was validated with a set too or another questionnaire? if that is the case please mention the correlation values.

Include information on questionnaire reliability testing, internal consistency, inter-rater reliability, or intra-rater reliability...also mention the association values of operators and items.

Discussion:

Add strengths of your study, limitations, and future directions to work on by fellow researchers.

Conclusion:

revise conclusion as per objective, need to focus on validation part.

Author Response

Dear Editor of the Special Issue “Prevention and Management of Oral Healthcare” of Journal of Personalized Medicine, 

Dr. Chun-Ming Chen 

We are pleased with the opportunity to revise and resubmit our manuscript entitled "Self-Reported Measures of Periodontitis in a Portuguese Population: A Validation Study” JPM (ISSN 2075-4426).

The comments from the reviewers were important and thoroughly considered. According to your instructions, below you will find a point-by-point response to the comments along with our revision. All changes are shown in the manuscript text file with track changes.

We hope this revised manuscript will enable its further consideration. We are happy to consider further revisions and we thank you for your continued interest in our research.

REVIEWER COMMENTS

Reviewer #2 (Remarks to the Author):

The study is useful and can contribute to the assessment of periodontal disease in a Portuguese population. The study has powerful statistics. Please clarify the following points prior to further processing.

Title: if validation is used in the title, then it should reflect in the study objective, otherwise your study is a "survey-based study"

Our answer: We have reflected the validation purpose on the study objective. Now the aim reads as follows: “

Introduction: Add significance of the study in the last paragraph.

Our answer: We appreciate this suggestion. As per your recommendation we added the significance of the study in the last paragraph that now reads as follows: “To our view, assessing the validity of self-reported questions to Portuguese and attesting its predictive capacity will be an important add up to surveillance of periodontitis.””

Methods:

The section should have a separate heading on questionnaire validation"

Our answer: We appreciate this remark, however we have followed the Transparent Reporting of a multivariable prediction model for Individual Prognosis or Diagnosis (TRIPOD)  statement guideline for validation purposes. Therefore, the heading order has strictly followed this guideline for transparency and clarity. 

Please clarify which methods were adopted to validate the questionnaire i.e., face validity, content validity, construct validity, or criterion validity.

Our answer: We recognize that such validation strategies would be possible. However, we followed the strategy implemented by previous studies on this topic, namely Montero et al (2020), Cyrino et al (2011), Heaton et al (2017), Iwasaki et al (2021) and Saka-Herrán et al (2020).

The questionnaire was validated with a set too or another questionnaire? if that is the case please mention the correlation values.

Answer: This set of questions are not considered part of a questionnaire per se, but rather a set of questions aiming to predict a positive diagnosis of periodontitis. Consequently, studies of validation have been implementing a strategy of correlating each question and a group of questions (through logistic regression) against dichotomous predictive results (to have or not periodontitis). In this study, the questions were validated against a defined classification of periodontitis (EFP 2018 or CDC/AAP 2012). The correlation values were presented in Table 3. We appreciate this commentary.

Include information on questionnaire reliability testing, internal consistency, inter-rater reliability, or intra-rater reliability...also mention the association values of operators and items.

Answer: Due to the specifications explained in the previous remark, and considering the purpose of this self-reported questions, we followed an equal strategy from previous studies on this topic, namely Montero et al (2020), Cyrino et al (2011), Heaton et al (2017), Iwasaki et al (2021) and Saka-Herrán et al (2020).

Discussion: Add strengths of your study, limitations, and future directions to work on by fellow researchers.

Our answer: We had already included a section focusing on strengths and limitations, as well future directions in the original submission that can be found in lines 260-271.

Conclusion: Revise conclusion as per objective, need to focus on validation part.

Our answer: We followed this reviewer's suggestion to provide a better description of the objective in the conclusion by saying ”Self-report measures of periodontitis combined with risk factors showed predictive validity towards periodontitis and its severe form in the Portuguese population using the 2018 and 2012 case definitions.”

References

Montero E, La Rosa M, Montanya E, Calle-Pascual AL, Genco RJ, Sanz M, Herrera D. Validation of self-reported measures of periodontitis in a Spanish Population. J Periodontal Res. 2020 Jun;55(3):400-409. doi: 10.1111/jre.12724. Epub 2019 Dec 24. PMID: 31872881.

Cyrino RM, Miranda Cota LO, Pereira Lages EJ, Bastos Lages EM, Costa FO. Evaluation of self-reported measures for prediction of periodontitis in a sample of Brazilians. J Periodontol. 2011 Dec;82(12):1693-704. doi: 10.1902/jop.2011.110015. Epub 2011 May 12. PMID: 21563951.

Heaton B, Gordon NB, Garcia RI, Rosenberg L, Rich S, Fox MP, Cozier YC. A Clinical Validation of Self-Reported Periodontitis Among Participants in the Black Women's Health Study. J Periodontol. 2017 Jun;88(6):582-592. doi: 10.1902/jop.2017.160678. Epub 2017 Jan 15. PMID: 28088874; PMCID: PMC5556388.

Iwasaki M, Usui M, Ariyoshi W, Nakashima K, Nagai-Yoshioka Y, Inoue M, Kobayashi K, Borgnakke WS, Taylor GW, Nishihara T. Validation of a self-report questionnaire for periodontitis in a Japanese population. Sci Rep. 2021 Jul 23;11(1):15078. doi: 10.1038/s41598-021-93965-4. Erratum in: Sci Rep. 2021 Aug 30;11(1):17673. PMID: 34301979; PMCID: PMC8302714.

Saka-Herrán C, Jané-Salas E, González-Navarro B, Estrugo-Devesa A, López-López J. Validity of a self-reported questionnaire for periodontitis in Spanish population. J Periodontol. 2020 Jan 26. doi: 10.1002/JPER.19-0604. Epub ahead of print. PMID: 31984491